# Comparative Root Transcriptome Profiling and Gene Regulatory Network Analysis between Eastern and Western Carrot (*Daucus carota* L.) Cultivars Reveals Candidate Genes for Vascular Tissue Patterning

**DOI:** 10.3390/plants12193449

**Published:** 2023-09-30

**Authors:** Chaitra C. Kulkarni, Sarvamangala S. Cholin, Akhilesh K. Bajpai, Gabrijel Ondrasek, R. K. Mesta, Santosha Rathod, H. B. Patil

**Affiliations:** 1Plant Molecular Biology Lab (DBT-BIOCARe), Department of Biotechnology & Crop Improvement, College of Horticulture, University of Horticultural Sciences, Bagalkot 587103, Karnataka, India; kchaitra642@gmail.com; 2Kittur Rani Chennamma College of Horticulture, Arabhavi, Gokak 591218, Belgaum Dt., Karnataka, India; 3University of Horticultural Sciences, Bagalkot 587103, Karnataka, India; 4Shodhaka Life Sciences Pvt. Ltd., Electronic City, Phase-I, Bengaluru 560100, Karnataka, India; 5Department of Soil Amelioration, Faculty of Agriculture, University of Zagreb, 10000 Zagreb, Croatia; 6Indian Institute of Rice Research, Hyderabad 500030, Telangana, India

**Keywords:** carrot, vascular cambium, candidate genes, gene regulatory network, plant stress response

## Abstract

Carrot (*Daucus carota* L.) is a highly consumed vegetable rich in carotenoids, known for their potent antioxidant, anti-inflammatory, and immune-protecting properties. While genetic and molecular studies have largely focused on wild and Western carrot cultivars (cvs), little is known about the evolutionary interactions between closely related Eastern and Western cvs. In this study, we conducted comparative transcriptome profiling of root tissues from Eastern (UHSBC-23-1) and Western (UHSBC-100) carrot cv. to better understand differentially expressed genes (DEGs) associated with storage root development and vascular cambium (VC) tissue patterning. Through reference-guided TopHat mapping, we achieved an average mapping rate of 73.87% and identified a total of 3544 DEGs (*p* < 0.05). Functional annotation and gene ontology classification revealed 97 functional categories, including 33 biological processes, 19 cellular components, 45 metabolic processes, and 26 KEGG pathways. Notably, Eastern cv. exhibited enrichment in cell wall, plant-pathogen interaction, and signal transduction terms, while Western cv. showed dominance in photosynthesis, metabolic process, and carbon metabolism terms. Moreover, constructed gene regulatory network (GRN) for both cvs. obtained orthologs with 1222 VC-responsive genes of *Arabidopsis thaliana*. In Western cv, GRN revealed VC-responsive gene clusters primarily associated with photosynthetic processes and carbon metabolism. In contrast, Eastern cv. exhibited a higher number of stress-responsive genes, and transcription factors (e.g., MYB15, WRKY46, AP2/ERF TF connected via signaling pathways with NAC036) were identified as master regulators of xylem vessel differentiation and secondary cell wall thickening. By elucidating the comparative transcriptome profiles of Eastern and Western cvs. for the first time, our study provides valuable insights into the differentially expressed genes involved in root development and VC tissue patterning. The identification of key regulatory genes and their roles in these processes represents a significant advancement in our understanding of the evolutionary relations and molecular mechanisms underlying secondary growth of carrot and regulation by vascular cambium.

## 1. Introduction

Carrot (*Daucus carota* L.) is widely cultivated spp. from *Apiaceae* family [1], with a diploid chromosome number of 2*n* = 2*x* = 18 and an estimated genome size of 473 Mb [2]. This crop holds a prominent position among primary vegetables, ranking in the top ten alongside tomato, onion, cabbage, cucumbers, and eggplant [3,4,5]. Moreover, carrot is recognized as the second-most important vegetable crop after potatoes, due to its nutritional composition and economic significance [6,7]. Over the past five decades, the global demand for carrots has steadily increased, with substantial growth in cultivated area (1.1 Mha) and production (40.0 Mt) [4]. Carrots are gaining popularity as a source of carbohydrates, minerals and vitamins, notably provitamin A [8]. In addition, carrots are a valuable source of carotenoids, important bioactive compounds that contribute to human health by offering a multitude of benefits, including antioxidant and anti-inflammatory properties, as well as immune and cardiovascular protection [9,10].

Considering morphological traits, cultivated carrots are classified into two types/cultivars: (1) Eastern, which include yellow, purple, or pale orange carrots, and (2) Western, primarily consisting of white, deeply pigmented orange or red carrots [8,11]. This classification is supported by genomics and transcriptomics studies as well [11,12,13,14,15,16,17]. Importantly, it was suggested that both carrot types have ancestral relationships with the domestication history [18]. For instance, Iorizzo et al. [2,17] suggest that Eastern carrot cvs. originated approximately 1100 years ago from wild populations in South Asia. The development of the modern dark orange Western carrot cvs. involved secondary domestication from the natural hybridization between yellow Eastern carrots and white-rooted accessions of wild carrots (*Daucus carota* L. ssp. *Carota*) [6,19]. Selective breeding of Western carrots has led to the enhancement of desirable characteristics, including a higher content of provitamin A and carbohydrates in uniform, juicy and smooth storage roots. However, Western carrot varieties exhibit a biennial flowering habit and require vernalization for optimal growth and development [11,18,19]. In contrast, Eastern carrot cvs. exhibit uneven shapes, larger core sizes, lower provitamin A content, coarser texture, and annual flowering, with minimal or no vernalization requirement for flowering [8].

Even though many genetic studies support a Central Asian origin of carrot domestication, specific molecular mechanisms driving domestication and underlying genotypic variability responsible for phenotypic changes, still remain poorly understood. While breeders have capitalized on the extensive phenotypic diversity present in carrots [20], a comprehensive investigation into the genotypic basis of these traits is still warranted [2]. Previous findings by Machaj et al. [21] provided insight into storage root development and genetic complexity between wild accession of *D. carota* subsp. *Commutatus* and domesticated orange-rooted Western carrots (*Daucus carota* ssp. *Sativus* var. *sativus*). However, there are still few comparative studies examining both the morphological and molecular aspects of these two carrot types. Namely, in carrots, the highest diversity has been found related to root thickening and vascular tissue differentiation, leading to genetic variation in root shape, color, length, and yield. In rooty crops, the development of storage roots and their yields primarily depends on the vascular cambium (VC) tissue. The VC acts as the key regulator, determining the thickening of xylem (internal) and phloem (external) tissues during the formation of secondary storage roots [22,23]. Continuous cell division (i.e., anticlinal type of cell division in VC) provides precursor cells for the secondary growth of the xylem inside the VC and phloem extension outside the VC by their radial growth [24]. In this process, several stress-responsive transcription factors (TF), VC and carbon partitioning genes play a crucial role. In coordination with environmental changes, a dynamic and complex gene regulatory network (GRN) operates in the VC for xylem and phloem formation [24]. In addition, cultivated plant species are exposed to a variety of abiotic [25] and biotic stresses [26], with negative impacts on their growth performances [27] and evolution [28]. However, it is believed that plant growth has an antagonistic relationship with stress resistance. Moreover, plants can prioritize their defense responses for survival by inducing stress-responsive gene cascades and suppressing growth-promoting regulators at the molecular level [29,30]. The molecular mechanisms underlying stress responses and growth modulation have widely been investigated in *Arabidopsis thaliana*, a model plant in stress biology [31], shedding light on the intricate balance between growth and stress adaptation [32].

In this study, we focused on analyzing the transcriptome of the storage root system in two important carrot cultivars: Eastern (*Daucus carota* ssp. *sativus* var. *atrorubens* Alef.) and Western (*Daucus carota* ssp. *sativus* var. *sativus*), with distinct domestication histories and valuable genetic resources for further investigation. In addition, GRN constructed with the VC regulatory genes separately for Eastern and Western carrot cvs. was compared with *Arabidopsis thaliana* from recent studies [24,33]. As a result, identified novel regulators of VC activity in response to growth and stress could (i) accelerate the genetic improvement of the carrot taproot, as well as (ii) elucidate molecular regulatory mechanisms governing modulation of key genes for stress resistance and growth.

## 2. Results and Discussion

### 2.1. Carrot Root Yield Depends on the Secondary Growth

A wide range of root plasticity parameters were recorded in the test carrot cultivars and are presented in Table 1. Eastern cv. has contrasting domestication traits with the Western cv. (Table 1) in terms of root shape, xylem-phloem patterning, growth, and flowering habit, which is in line with some previous studies [8,11]. Significant variations between test cvs. were found in morphometric parameters such as root yield, shoulder width, xylem and phloem color and size, carotenoid accumulation, growth habit, and blooming behavior (Table 1). Western cv. obtained the highest phloem width (86 mm) with uniform pigmentation and little xylem patterning (40 mm), whereas Eastern cv. showed the highest xylem width (84 mm), but lower phloem width (50 mm) (Table 1). Uniform exterior and interior colors are important characteristics receiving attention in carrot breeding programs. The average shoot weight (70.91 ± 12 g) of Eastern cv. was higher in comparison to Western cv. (24.44 ± 3.46 g). However, average root weight of Western cv. (40.66 ± 4.11 g) was lower than in Eastern cv. (72.70 ± 5.88 g). Western cv. exhibited significantly higher β-carotene content (8.5 mg/100 mg) than Eastern cv. (6.5 mg/100 mg; Table 1). The total soluble solids (TSS) varied from 6.86° Brix in Western cv. to 7.08° Brix in Eastern cv. These morphometric variations are useful tools in the (pre)selection of desirable cvs. [34].

In addition, the average root width showed a strong positive correlation with average root weight (R^2^ = 0.675) (Table 2). The vegetative weight has a positive correlation with xylem width (R^2^ = 0.440); however, it has a negative correlation with phloem width (R^2^ = −0.166). The root width was influenced positively by xylem (R^2^ = 0.432) and phloem width (R^2^ = 0.44). As shown by a strong correlation, the xylem width will increase the root weight (R^2^ = 0.726) markedly more than the phloem width (R^2^ = 0.342) since the xylem core is harder and heavier. Both the xylem and phloem obtained a positive correlation with root weight and root width, indicating that radial root growth is an important factor determining root yield in carrots. An increase in shoulder width is correlated with higher xylem width (R^2^ = 0.869) than phloem width (R^2^ = 0.450). Hence, Eastern cv. Has a higher root weight than Western cv. (Table 1); however, the energy is mostly diverted to the shoulder portion and to the xylem core formation, making it coarser in texture. The poor negative correlation between shoot length and root width (R^2^ = −0.091) and phloem width (R^2^ = −0.115) was recorded. In addition, phloem width had a negative association with plant vegetative weight (R^2^ = −0.166), number of petioles (R^2^ = −0.178), shoot length (R^2^ = −0.115), and root length (R^2^ = −0.059), whereas xylem width had significant positive correlations with plant height (R^2^ = 0.692), petiole length (R^2^ = 0.432), root width (R^2^ = 0.726), and vegetative weight (R^2^ = 0.440) (Table 2). This result highlights the importance of achieving a uniform phloem pattern in carrots, which is associated with lower vegetative weight, shorter petiole length, and reduced shoulder width. Western cvs. Evolved with these desirable characteristics through careful selection during domestication [19], whereas Eastern cv. Had limited exposure to the domestication of such traits. Therefore, to gain further insights into the genetic basis of secondary root growth and VC patterning in Eastern and Western carrot cvs., transcriptome profiling was employed to explore genes and regulatory pathways involved.

### 2.2. Reference-Based Assembly and DEGs in Storage Root Transcriptome

RNAseq technology was employed in the current study to retain 130.62 M (92.71%) high-quality (Phred score > 30), cleaned paired-end reads out of a total of 140.88 M reads. The cleaned reads from each library, with the range of 17.63 M to 24.93 M reads per sample, were mapped using reference-based assembly (Table 3), resulting in more reads than previous studies on the transcriptomics of carrots [21,35] or other storage rooty crops such as radishes [36]. The TopHat assembly mapping rate varied from 71.6 to 82.8%, with an average mapping percentage of 73.87%. Mapping rates in Western cv. ranged from 71.6 to 82.8% and in Eastern cv. from 68.5 to 72.7% (Table 3), which is markedly higher than in some similar studies, e.g., from 0.06 to 66.61%, [37].

A total of 3544 DEGs were identified by comparing expression levels of genes or other genomic features between test carrot cvs. using storage root transcriptome analysis. In Eastern cv., 1817 genes were up-regulated, while 1727 genes were found to be up-regulated in Western cv. (Figure 1a,b). Conversely, genes that are up-regulated in Eastern cv. are down-regulated in Western cv., and vice versa. Gene expression coverage across the chromosomal regions is a useful feature for analyzing DEGs spread across the genome (Figure 1c). Difference in DEGs between tested cultivars reflects the possible molecular mechanisms that control storage root development and VC patterning or their significance in domestication and evolution [6,19].

### 2.3. Functional Annotation of DEGs

In the DAVID v6.8 web server, using *Daucus carota* subsp. *sativus* L. as a reference, a total of 2776 (78.33%) of the 3544 DEGs in the root transcriptome were functionally annotated. Among these, 70.83% (1287) and 86.21% (1489) of the 1817 up-regulated genes in Eastern cv. (Appendix A) and 1727 up-regulated genes in Western cv. (Appendix A), respectively, were successfully annotated. The remaining 768 genes were either uncharacterized or had no function assigned. The up-regulated DEGs in Eastern cv. were associated with “innate and adaptive immune response” inclusive of WRKY transcription factors, ABC transporters, and Mitogen-activated protein Kinase (MAPK) signaling. Furthermore, cellular process (cell cycle checkpoint protein, cell wall protein, cellulose synthase-like protein D3, cell division control protein, xyloglucan endotransglucosylase), cell wall synthesis (cell wall protein RBR3, repetitive proline-rich cell wall, pectin esterase), pathogenesis-related genes, phenylpropanoid, Ca signaling, and various chaperon proteins were also up-regulated (Appendix A). Likewise, cytochrome P450-like, ubiquitin-protein ligase, gibberellin-regulated protein, kinesin-like protein (KIN-like), leaf rust disease resistance, leucine-rich receptor protein-related DEGs were higher in the Eastern cultivar. Heavy metal-associated isoprenylated plant proteins (HIPPs) that are involved in heavy metal homeostasis and detoxification in cells, serine-threonine protein kinase, NAC, WRKY TF, and U-box domain-containing protein, had higher accumulation in Eastern cv. In addition, various plant hormones responsive transcripts (auxin-responsive protein-SAUR32, IAA26, auxin response factors 3, 4, auxin-induced proteins, cytokinin dehydrogenase 7-like, gibberellin-regulated protein, ethylene-responsive transcription factors-ERF-061, 014, 25, CRF4-like, abscisic acid-insensitive, etc.), NRT1/PTR gene family in response to salt stress, etc., were highest in Eastern cv. that is necessary for wider adaptation to climate change. A total of 58 TF of different classes that play key roles in gene expression were found in Eastern cv. (Appendix A). A significantly higher number of protein kinases (104) was detected in Eastern vs. Western cv. (36) (Appendix A). Throughout the plant cycle, root growth during biotic stress responses and MAPK signals are extremely important [36,38,39,40]. Phenylpropanoid biosynthesis is also important in releasing the phenols against the water stress and wound stress in carrots, which helps in pre-/post-harvest management [41].

Possible selection signatures are identified by the higher accumulation of photosynthesis-related genes (photosystem-I reaction center and photosystem II protein, photosynthetic NDH subunit), carotenoid biosynthetic pathway genes (phytoene synthase, phytoene desaturase), and chlorophyll a-b binding proteins were more highly accumulated in Western cv. than in Eastern cv. (Appendix A). A higher accumulation of carotenoids is essential for photosynthesis and has previously shown prominent diversity among the yellow, red, and orange carrot roots [5]. Our results with Western cv. are also supported by a carrot genome study [2] showing the upregulation of photosynthetic pathway genes and Photosystem-I and II in orange carrots. Photosynthates reach roots by sieve components, companion cells, and plasmodesmata to facilitate secondary root development [42,43]. The uniform dark orange storage root in Western cv. is associated with photosynthetic efficiency [2,5] and could be a rate-limiting pathway in breeding for higher carotenoid accumulation in Eastern cvs. In addition, heat shock proteins, auxin-responsive proteins, thylakoid luminal protein, F-box protein, F-box-Kelch repeat protein were found to be higher in Western cv. Similarly, Dof zinc finger protein, MYB TF, probable WRKY TF, oxygen-evolving enhancer protein, Zn finger proteins and pentatricopeptide repeat-containing protein that facilitates processing, splicing, editing, stability and translation of RNAs [44] were also found to be higher in Western cv. (Appendix A).

Important transcription factors such as ERF, heat stress TF, NAC-29 TF, WRKY, bHLH, CYCLOIDEA TF, DIVARICATA, MYB108, MYC2, RAX2, TCP4, trihelix TF GT-3b-like, bZIP, AP2/ERF, ethylene-responsive TF-CRF4 were up-regulated in Eastern cv. (Appendix A). These TF play an important role in the regulation of storage root development, defense mechanisms, signal transduction, and abiotic stress tolerance [21]. Parallel, in Western cv., TF such as bHLH, DIVARICATA, MYB, E2F, HBP-1b, PCL1, PIF7, SPATULA, TCP, TGA, VIP-1 like, RAV-1 like, B3 domain containing VRN1 like, ERF, RAP2, TINY-like, NF-YA, GATA, and WRKY were up-regulated (Appendix A). PIF TF regulates photomorphogenesis in carrot, plays an important role in hypocotyl elongation in the dark, and enhances carotenoid accumulation [45]. VRN1 plays a crucial role during vernalization to repress the FLC gene for a successful vegetative to reproductive transition [46]. The divergence of the type and class of transcriptional factors between test cultivars would be the result of selection signatures of two domestication events in carrots and play a possible role in the differential expression of domesticated traits, especially regarding root quality in terms of carotenoid accumulation and defense response to various stresses.

Protein kinases are a class of proteins important in post-translation modification by phosphorylation, protein conformation, stability, localization, signal transduction, and cell regulation, and responses to various environmental stresses [47]. Interestingly, various classes of protein kinases were found to be higher in Eastern cv. (104; Appendix A) than in Western cv. (only 36; Appendix A), which can be interrelated to a higher accumulation of signaling pathways in Eastern cv. Furthermore, Machaj et al. [21] observed a significant increase in protein kinase accumulation during the storage root development of the wild carrot relative to Western cv.

### 2.4. Gene Ontology (GO) Terms and KEGG Pathway Annotation of Differentially Expressed Genes

To further establish the biological function of DEGs responsible for storage root development and VC patterning in carrots, the functional annotation was performed by mapping all DEGs to Gene Ontology (GO). A total of 3544 DEGs in the root transcriptome were classified into 97 functional groups consisting of 33 biological processes (BP), 19 cellular processes (CC), and 45 molecular functions (MF) subcategories (Figure 2a–c & Appendix A). The BP terms in Eastern cv includes phosphate ion transport, metabolic processes (sucrose metabolism, fructose metabolism, xyloglucan), cell wall organization, cell wall biogenesis, microtubule-based movement, and other processes. The structure of the cell wall plays critical functions in plant development, cell differentiation, cell expansion, intercellular communication, and defense [48]. CC terms were related to integral components of membrane, cytosol, plasma membrane, plasmodesmata, and cell wall specifying nuclear and inter-cellular transport and localization. MF terms were shared with ATP binding, oxidoreductase activity, protein serine/threonine kinase activity, xyloglucan transferase sequence-specific DNA binding, and others. This GO analysis was implemented with Bonferroni corrected FDR ≤ 0.01 and fold enrichment 1.0-11.51 (Appendix A). In the KEGG, important signaling pathways (MAPK signaling, phenyl propanoid pathway), metabolic processes, ABC transporters, and plant-pathogen interaction enrichment in Eastern cv. support the strong root development that has wider buffering capacity and climatic adaptation to various (a)biotic stresses (Figure 2d). Plants have signal transduction pathways, which are complex networks of interactions involving signal elements transmitting through the plant cell and allowing them to respond appropriately to a specific environmental stimulus. Cell signaling influences nearly every aspect of plant cell structure and function [49]. The absence of these GO terms during the subsequent secondary domestication event in Western carrots would be hypothesized as a domestication bottleneck.

Conversely, up-regulated BP in Western cv. was related to the regulation of transcription, photosynthesis (photosystem-I, photosystem-II, oxygen-evolving complex), metabolic process (lipid metabolism, carbon metabolism, biosynthesis of secondary metabolites), and defense response (Figure 2a–c & Appendix A). Genes in photosynthesis were variably expressed and substantially enriched during storage root formation in cultivated orange carrots and repressed in wild carrots in previous comparative transcriptomics research [21]. The carotenoid pathway is enriched in early or developing roots, whereas in adult roots, carotenoid biosynthesis genes are less common and co-expressed with chloroplast genes [21]. Increased carotenoid accumulation in mature carrot roots is correlated with greater expression of genes involved in photosynthesis, such as LHC-II [2]. CC terms in Western cv. were confined to the nucleus, chloroplast thylakoid membrane, photosystem II, photosystem I, oxygen-evolving complex, photosystem I reaction center, chloroplast thylakoid lumen, and nucleus. MF expressions in Western cultivars were DNA binding, metal binding, transcription factor activity, and chlorophyll binding (Appendix A). In carrots, genes involved in plastid formation, photosystem I and II assembly, and carotenoid biosynthesis are co-expressed with isoprenoid pathways [2,21,50]. In summary, extensive analysis has demonstrated that the gene expression patterns and enriched pathways in Eastern and Western cvs. are unequivocally distinct from each other.

To further identify the differentially expressed transcripts that are involved in the same or different biological pathways and to gain deeper insight into the enriched pathways between test carrot cultivars, KEGG pathway analysis was carried out to emphasize the network of gene products (Figure 3a,b). It was found that metabolic pathways (200 genes) and biosynthesis of secondary metabolites (120), plant-pathogen interaction (29), phenylpropanoid biosynthesis (21), MAPK signaling, and ABC transporters were up-regulated pathways in Eastern cv. (Figure 2d and Appendix A). For Western cv., metabolic pathways (209 genes) were followed by photosynthesis (40), carbon metabolism (25), plant hormone signal transduction (24), along with six other KEGG pathways (Figure 2d and Appendix A). This shows the systematic evolution of the photosynthetic pathway for higher carotenoid accumulation in Western cv. as supported by previous findings [2,5] and our KEGG pathway enrichment analysis (Figure 3b). Further KEGG pathway enrichment analysis confirmed the enrichment of pathways viz., “alpha-linoleic acid metabolism”, “ABC transporters”, “plant-pathogen interaction”, “phenylpropanoid pathways”, ‘metabolic processes’ in Eastern cv. (Figure 3a), and “metabolic processes” followed by “photosynthetic pathways” and “carbon metabolism” in Western cv. (Figure 3b). These results suggest that differentially expressed transcripts likely have a significant impact on the development of secondary storage roots and the transportation of photosynthates from leaves to roots but in distinct patterns between tested cvs.

#### 2.4.1. DEGs of Cell Cycle and Cell Wall Metabolism Transcript Profiles

Genes associated with the cell cycle were observed as differentially regulated in carrot storage root (Appendix A). For instance, cell division control protein 2 (CDC2), cyclin-dependent kinase (CDKs), cyclin-D1-1-like, B2-1-like, and cyclin-dependent kinase inhibitor (CDKI) 4-like SMR were up-regulated in Eastern cv. In addition, glycine-rich cell wall structural protein, putative cell division cycle ATPase, repetitive proline-rich cell wall protein, programmed cell death protein, and accelerated cell death related DEGs were up-regulated in Eastern carrot (Appendix A). The cell wall not only strengthens the plant body, but also has key roles in plant growth, cell differentiation, cell expansion, intercellular communication, water transport, and defense [51,52].

In the present study, transcripts encoding key enzymes that are involved in the cell wall synthesis and degradation, such as cellulose synthase, galactosyltransferase, UDP-glycosyltransferase, and caffeoyl-CoA O-methyltransferase, endoglucanase, pectin acetyl esterase, pectin esterase, Xyloglucan endotransglucosylase protein-like-related DEGs were up-regulated during Eastern secondary storage root development compared to Western carrot (Appendix A). The cell cycle and division of cells determine the size of cells and organs [53]. Interestingly, higher numbers of transcripts homologous to cell division and cell wall metabolism were associated with Eastern carrot cv. (Figure 4a).

#### 2.4.2. DEGs with Signal Transduction and Related Transcriptome Profiles

The mitogen-activated protein kinase (MAPK) was reported to play a key role in regulating the cell cycle and developmental processes [35]. Two MAPK (MAPK-9 and NPK-1-like) were found in Western cv. (Appendix A), and four (MAPK-3, 9, MMK2 and NTF6) were found in Eastern cv. (Appendix A). In addition, KEGG has also been enriched with MAPK signaling pathway in Eastern cv. (Appendix A). Ca-regulated transduction pathway is another key signaling transduction category in cell development [35]. In Western cv., only two Ca-binding proteins (CML49 and CML36) were identified (Appendix A). However, in Eastern cv, eight Ca-binding proteins and calmodulin (CaM) proteins (CML30, CML18, CML49, CML20, CML25, CML45, CML23, CML24) were found (Appendix A). This suggests that pathways involved in Ca transport are more extensive in tested Eastern cv. than in Western cv. (Appendix A). Furthermore, the up-regulation of various genes in Eastern cv. (Appendix A) indicates their potential role in important physiological processes. These include ABC transporters, WRKY transcription factors, plant pathogen interacting respiratory burst oxidase protein (RbohD) responsible for signaling reactive oxygen species (ROS), and caffeoyl-CoA O-methyltransferase-like enzymes involved in the phenylpropanoid pathway (Figure 4b, Appendix A).

#### 2.4.3. DEGs of Starch and Sucrose Metabolism Transcript Profiles

Carbon-related metabolism genes (e.g., beta-amylase, sucrose synthase, sucrose phosphate synthase (SPS), UDP-glucose transferase, glucose phosphate, glucose dehydrogenase, beta-glucosidase, beta-galactosidase, beta-fructofuranosidase, and beta-xylosidase) as suggested by Treves et al. [54] exhibited down-regulation in Western cv. in contrast to Eastern cv. (Figure 4c, Appendix A), suggesting distinct differences in carbon metabolism and accumulation between test cultivars. However, in Western cv., KEGG analysis revealed a higher enrichment in metabolic pathways (vs. signal transduction pathways) such as photosynthesis, carbon fixation, glycolysis, gluconeogenesis, and carbon metabolism (Figure 3b). In addition, in Eastern cv., synthesis of higher levels of carbohydrates may also be involved in various signal transduction pathways and other biological processes, which is evident from KEGG analysis (Figure 3a). Higher accumulation of sucrose is necessary during storage root development [55]. These results suggest that many key functional genes are differing in the storage root between test carrot cultivars. Therefore, the GRN was constructed to find the predominant active pathway in the storage root of carrot cultivars and master regulators involved in the swapping of root growth and stress response (Figure 4c).

### 2.5. GRN Associated with VC Patterning between Test Carrot Cultivars

The GRN analysis revealed a higher number of stress-responsive genes and transcription factors (LOX2, LOX3, LOX4, WRKY46, MYB15, ERF4, PUB24, KIN3, NAC036, NLA etc.) in Eastern (vs Western) cultivars, along with a few genes related to root growth (PERK10, AT2G26380 Leucine-rich repeat (LRR, LSH6, HCA2) and carbohydrate metabolism (NANA)), with no significant impact on photosynthesis network genes in Eastern cv. (Figure 5a, Appendix A). In addition, GO enrichment analysis highlighted 83 biological process (BP) terms, 12 molecular function (MF) terms, 7 cellular component (CC) terms, and 10 KEGG pathway enrichments (Appendix A) associated with VC-responsive genes specific to Eastern cv. Several of the target genes in GRN were implicated in cell cycle events, plant hormone regulation, and various signal transduction and metabolic processes, including cell division, differentiation, expansion, auxin signaling, sucrose metabolism, and energy metabolism (Figure 5a). These findings suggest their crucial role in taproot thickening during carrot development.

VC-responsive genes crucially play a role in maintaining the uniform xylem phloem pattering with the help of various regulatory mechanisms. In Eastern cv. XTH 7, XTH9, SVB, PhyA, HCA2, HB-12, and NAC036 are abundantly expressed in cambial regions during the secondary root growth (Figure 5a). These not only regulate xylem cell expansion but also influence several characteristics of secondary growth, including secondary xylem production and secondary wall deposition [56]. HCA2 is a Dof-type zinc finger DNA-binding family protein, which induces the formation of interfascicular cambium and regulates VC tissue development [24]. NAC036 present in Eastern cv. was identified as the master regulator of xylem vessel differentiation and secondary cell wall thickening [57]. WOX4 (AT1G46480) present in Eastern cv. promotes differentiation and/or maintenance of the vascular procambium. Down-regulation of WOX4 in *Arabidopsis* generated small plants with undifferentiated ground tissue and exhibited severe reductions in differentiated xylem and phloem [58]. The genome editing approach for this gene would facilitate uniform patterning in Eastern carrot cvs. Integration of DEGs and gene interactions in GRN provided insights into the intricate regulatory network associated with carrot taproot development, thickening, and VC patterning.

In Western cv., there was an up-regulation of photosynthetic cascades and photosynthesis-related genes, as observed in GO subcategories of BP and CC (Figure 5b, Appendix A). Similarly, the GRN using up-regulated VC-responsive genes in Western cv. revealed the clustering of genes involved in photosynthetic activities (ex: LSH10, PSAH, PSAN, PSAD, LHCB, PSB, etc.), along with a few stress-responsive genes (WRKY70, ERF-71, MKK9, MYB15, CAMBP25) and root vascular developmental genes (TRN2, LSH10, SP1L2, HB6, AT1G66830) (Figure 5b, Appendix A). GO enrichment analysis of 109 up-regulated DEGs in Western cv. identified 82 enriched terms in BP, 18 in MF, 33 in CC, and 6 enriched KEGG pathways (Appendix A) related to the VC-responsive genes (Appendix A). Such results show that VC-responsive genes were co-expressed with photosynthetic processes leading to higher and more uniform carotenoid accumulation. It is well known that Western carrot cvs. have reduced genetic diversity [11,17] with increased desirable traits during domestication [8] and are highly susceptible to various pathogens [8,59]. This is also supported by our results because the root development and vascular tissue development utilizes higher photosynthates instead of diversion to other stress-responsive pathways, as confirmed in Eastern cv. TRN2 regulation is required for initial meristematic divisions in the epidermal layer and to maintain the radial phloem pattern of cell specification in the root [60]. LSH10 is a developmental regulator involved in deeper carbohydrate assimilation in storage roots [61]. AT1G66830 is an LRR-protein kinase found as a master regulator that cooperates with phloem intercalated XYLEM (PXY)/TDIF receptor, promoting cell divisions in the cambium, inhibiting xylem differentiation, and controlling VC patterning [62]. Several edible secondary storage rooty crops, such as potato, radish, yam, cassava, and sweet potato, have comparable results [24]. The inclusion of these two cultivars in a genomic-assisted quality improvement program could help in (i) the development of climate-resilient Western carrot cvs, and (ii) higher carotenoid accumulation with uniform ‘self-core’ and high-quality roots in Eastern carrot types.

Root development and response to the environment are controlled by the GRN [63]. To further understand the enhanced regulatory network pathways between test cvs, GRN was built utilizing DEGs of our root transcriptome orthologous to VC of *Arabidopsis*. The availability of a higher number of homologous genes with the major evolutionary closeness between carrot and *Arabidopsis* unfolds the major regulatory genes in this comparative investigation [64,65,66,67]. Signal transduction pathways, such as those involved in hormone, calcium, and MAPK signaling, as well as metabolic processes related to cell wall, carbohydrates, storage, and energy metabolism, play crucial roles in energy and nutrient absorption by cells. Regulatory elements influence the differentiation, division, and expansion of secondary xylem and phloem in root development [24]. To devise strategies for sustainable yield in rooty crops, it is essential to understand the cambium regulatory programs that drive storage root development.

### 2.6. Expression Validation of DEGs by qPCR

A total of 9 DEGs were randomly chosen and validated by RT-qPCR analysis to assess the validity and reproducibility of our DEG findings (Appendix A). For each sample, the Log_2_ ratio of the FPKM value was utilized to reflect the fold changes in comparison to the Log_2_FC estimated based on the CT value of qPCR (Figure 6). For eight of the nine genes (G1 to G9), the fold change values from RNAseq and qPCR showed a concordant tendency of gene expression. Eight genes matched the direction of the change fold. The expression of G1 was discordantly expressed in opposing directions by RNAseq and qPCR. Current annotation indicates that the root growth is regulated by the genes: G1 (Mechano-sensitive ion channel protein 6-like), G2 (ubiquitin-conjugating enzyme E2 20-like; LOC108208449), G3 (polyubiquitin; LOC108227972), and G4 (probable WRKY transcription factor53; LOC108209169). Then, G6 (cysteine proteinase inhibitor B; LOC108211283) is engaged in seed germination, while G5 (probable glycosyltransferase At5g03795; LOC108193533) is likely involved in carbon metabolism. The innate immune response involves G7 (ATPase WRNIP1; LOC108207021), G8 (DNA repair protein XRCC2 homolog; LOC108216678), and G9 (histone deacetylase 6; LOC108200546) (Plant pathogen interaction). G1 (mechanosensitive ion channel protein 6-like; LOC108226635) participates in signaling pathways and homeostasis, but their interactions with other genes are involved in growth and development and are sensitive to hormones, light, and stress. With a few exceptions, it was discovered that relative expression changes based on qRT-PCR results were consistent with the RNAseq data (Figure 6). Results obtained through RNA sequencing (RNA-seq) offer a higher degree of precision because this method provides comprehensive gene expression data at the genome-wide level, allowing for the quantification of gene expression levels across the entire genome. RNA-seq is considered reliable due to its reliance on a substantial number of biological replicates, usually three or more, which enhances its credibility. In contrast, qRT-PCR techniques are typically employed to measure the expression levels of a limited number of genes. While there are situations where this approach may offer advantages, it is worth noting that RNA-seq methodologies and data analysis procedures have reached a level of reliability that often renders the validation of results by qPCR or other methods unnecessary [68,69,70].

Our findings contribute to the improvement of Eastern cv. by identifying genes responsible for key domesticated properties such as enhanced carotenoids with consistent xylem-phloem color [11] and Western cv. improvement against different diseases and pests [8,55]. To our knowledge, this is the first report to identify likely candidate genes responsible for VC patterning in Eastern and Western carrot cvs. in secondary storage roots. The markers developed for the candidate genes of the present study are valuable for genetic diversity, genomic-assisted breeding, or genomic selection in carrot and related crop species [71,72].

## 3. Materials and Methods

### 3.1. Plant Material Preparation for Transcriptome Profiling

The vegetative phase of the study was conducted in Bagalkot, Karnataka, India (16°12′ N, 75°45′ E), during 2020 rabi season in a greenhouse. The average elevation in this area reaches approximately 610 m, with an average annual rainfall of 318 mm and a semi-arid tropical climate. For the storage root phenotypic evaluation and RNA sequencing, two carrot cultivars were selected: Eastern (*Daucus carota* ssp. *Sativus* var. *atrorubens* Alef) and Western (*Daucus carota* ssp. *Sativus* var. *sativus*) subgroup of the association panel [11]. Eastern cultivar UHSBC-23-1 is characterized by light orange pigmentation, clear vascular tissue differentiation, and non-uniformity in the xylem and phloem patterning. Western cultivar UHSBC-100 was a commercial dark orange ‘Kuruda’ type with a completely uniform xylem and phloem with a thin cambium line. Test cultivars were sown in sandy loam soil, with a pH between 6.0 to 7.0. At 90 days after sowing (DAS), fully matured and well-developed roots from both cultivars were harvested. For measuring various quantitative and qualitative traits related to root plasticity and other root morphometric characters, 100 plants were randomly selected from each cv. from the bulk. Quantitative traits such as plant height (cm), root length (cm), shoot length (cm), xylem width (mm), and phloem width (mm) were recorded with a measuring scale. Root width (cm) and shoulder width (cm) were measured using a digital vernier caliper. Root weight (g), shoot weight (g), and plot yield were measured using a digital weighing balance. Root-to-shoot ratios were estimated from the root length (cm) or weight (g) and shoot length (cm) or weight (g). Total soluble solids (TSS in °brix) were estimated using a refractometer, and β-carotene was estimated using the acetone method [73]. Qualitative parameters were recorded with the help of the minimum characterization descriptor of carrot [74]. The tendency of flowering was observed by growing the root shoulders in a polyhouse in Bagalkot, Karnataka, India. Eight weeks of *vernalization* was performed with a temperature of 0 to 4 °C. A Student t-test assuming equal variance was performed to find the significant differences between test cultivars for individual traits. Furthermore, Pearson’s correlation coefficient analysis was performed to understand the correlation between the quantitative traits using the software package PAST v4.03 [75]. For transcriptome profiling, to ensure accurate analysis, three biological replicates were collected, each replicate consisting of two roots from each cultivar. The harvested roots were promptly frozen in liquid nitrogen to preserve their molecular integrity. The cryopreserved samples were then transported in dry ice on the same day of collection to proceed with the sequencing process. 

### 3.2. RNA Isolation, Library Preparation, and Sequencing

Samples with RNA integrity number (RIN) > 8.0 were subjected to cDNA synthesis and library construction for RNAseq. cDNA libraries were prepared for transcriptome profiling using Hiseq Illumina 4000. Approximately 5–10 ug of total RNA was used to prepare the RNAseq library using the TrueSeq RNA sample preparation Kit (Illumina). In short, poly A-containing mRNA molecules were purified using poly-T oligo-attached magnetic beads. Following purification, the mRNA was fragmented using divalent cations under elevated temperatures. The cleaved RNA fragments were used to synthesize first-strand cDNA using reverse transcriptase and random primers followed by second-strand cDNA synthesis using DNA polymerase I and RNase H. These cDNA fragments underwent an end repair process, the addition of a single ‘A’ base, and then ligation of the adapters. The products were purified and enriched with PCR to create the final cDNA library. Bioanalyzer plots were used at every step to assess mRNA quality, enrichment success, fragmentation, and final library sizes. The size distribution of the sequencing library was determined by gel electrophoresis. Qubit (Q) was used for measuring the quantity of the library before sequencing. For sequencing, constructed libraries were processed on HiSeq 4000, resulting in paired-end (PE) reads of 2 × 100 base pairs. Each library generated a substantial number of reads, ranging from 25 to 30 M. Over 90% of the reads exhibited a Q value of 30 or higher, indicating high-quality sequencing data.

### 3.3. Reference Based Assembly & Functional Annotation of Differential Expressed Genes (DEG)

Raw reads of six paired-end libraries were passed through quality check using the FASTQC (Version 0.11.9) (https://www.bioinformatics.babraham.ac.uk/projects/fastqc/ accessed on 20 May 2021) and analyzed for parameters such as basic statistics, adapter content, sequence length distribution, sequence duplication level, per base sequence quality, and Guanine: Cytosine (GC) content. The raw data of six RNAseq libraries have been submitted to Sequence Read Archives (SRA) in the NCBI database under the project ID PRJNA913450. Depending on the errors reported by FASTQC, additional pre-processing steps were carried out. The adapter was trimmed for each sample individually by applying Trimmomatic v0.36 [76]. The parameters with min quality 30, min length 50, LEADING:28, TRAILING:28, SLIDINGWINDOW:10:28, MINLEN:50 was assigned. After filtering duplicate and low-quality reads, 130,618,433 (92.71%) high-quality cleaned reads remained out of 140,882,952. Bowtie 2 (v.2.4.4) [77] was used to get the index files of the carrot reference genome assembly (ASM162521v1-Refseq GCF_001625215.1) downloaded from ncbi.nlm.nih.gov accessed on 20 May 2021. TopHat [78] was used for mapping the individual libraries with the help of generated index files and reference assembly. A total of 2 mismatches and up to 3 bp *indels* were allowed in alignment. Mapped RNAseq was assembled with Cufflink’s suite v. 0.17.3 [79] for normalization and transcript quantification by Fragments Per Kilobase of exon per Million (FPKM) fragments mapped values. Differentially expressed gene (DEG) analysis was performed with the cuff-diff tool of the Cufflinks suite v. 0.17.3 [79].

Statistical significance (*p*-values) was estimated by computing the False Discovery Rate (FDR) using the Benjamini–Hochberg correction [80]. Differences in gene expression were statistically significant when the q-value (FDR-adjusted *p*-value) was <less than 0.01. The criteria considered for filtering significant DEG included Log_2_Fold Change (Log_2_FC) ≥ +1 as up-regulated in Eastern cv. and ≤−1 as up-regulated in Western cv. The list of DEGs with Entrez gene IDs was imported into DAVID 2021 bioinformatics resource [81]. The annotated genes and functionally classified gene clusters (based on enrichment score) were downloaded to understand the biological meaning of the DEGs. The up- (Log_2_FC ≥ 1) and down-regulated (Log_2_FC ≤ −1) DEGs were separately imported to DAVID and ShinyGo [82] for functional annotation and Gene Ontology (GO) analysis. GO categories such as Biological Processes (BP), cellular components (CC), MF (Molecular Functions), and KEGG (Kyoto Encyclopaedia of Genes and Genomes) pathways for up- and down-regulated genes were downloaded and compared with our results.

### 3.4. Gene Regulatory Network (GRN) Analysis with Arabidopsis thaliana Homologs

GRN analysis was carried out with the list of genes identified as VC-responsive genes in *Arabidopsis thaliana* (AT) [24,33]. The upregulated genes of test carrot cvs. were extracted from the DEGs and subjected to *Arabidospis thaliana* (AT) orthologs search in “g:orth” in g:Profiler web browser. In brief, a total of 1794 out of 3544 DEGs were successfully obtained with AT gene ID. Then, obtained AT gene IDs were searched for common genes in 1222 AT genes that are considered key VC regulatory genes [33]. The common genes from this study and by Zhang et al. [33] were further sorted for up-and down-regulated genes for corresponding test carrot cvs. DEGs. A total of 198 DEGs out of 1794 DEGs perfectly matched the list of VC regulatory genes of Zhang et al. [33]. A total of 89 DEGs up-regulated in Eastern cv. (Appendix A) and 109 DEGs up-regulated in Western cv. (Appendix A) were identified as VC regulatory genes/transcriptional factors for GRN construction.

### 3.5. Expression Validation of DEGs by qPCR

A total of nine genes identified as DEGs were selected for the validation of gene expression using quantitative real-time PCR. The most stably expressed reference gene “Actin” was used as an endogenous control in the root qPCR experiment to validate RNAseq target genes. The details of genes used for qPCR validation are presented in Appendix A. The relative expression levels of target genes were calculated using the 2^−ΔΔCt^ method [83] with Actin as an internal control. The RNA pools used in the qRT-PCR analyses were extracted from three independent samples from Eastern (UHSBC-23-1) and Western (UHSBC-100) carrot cvs. storage root tissue.

## 4. Conclusions

The present study provides a comprehensive understanding of the development of storage roots in Western carrot cv. by elucidating how carbon flow influences phenylpropanoid biosynthesis and stress response pathways, while also contributing to carbohydrate metabolism, starch biosynthesis, and secondary metabolite formation. These processes play crucial roles in the initiation and development of carrot storage roots during domestication and adaptation in ecologically distinct regions. The transcript data generated in the present study is valuable for global carrot breeders to understand the domestication genes between evolutionarily related test cvs.

The enrichment of DEGs associated with photosynthesis, carbon metabolism, and carotenoid-related genes in Western cv. underscores its potential for nutritious and uniformly deep-colored carrots. Conversely, Eastern cv. exhibits a significant enrichment in genes responding to various (a)biotic stresses, along with VC genes. For effective plant breeding programs, a comprehensive understanding of VC-responsive genes controlling xylem and phloem patterning pathways is essential. Consequently, genetic engineering of master regulators of VC patterning genes would bring a uniform xylem-phloem patterning in Eastern carrot cvs.

## Figures and Tables

**Figure 1 plants-12-03449-f001:**
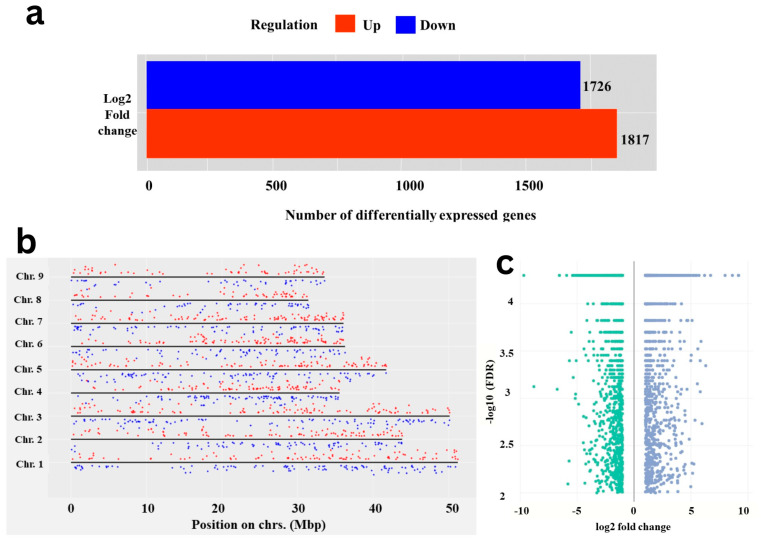
Differentially expressed genes in Eastern and Western cultivars root transcriptome. (**a**) Bar plot representing the total number of up- and down-regulated DEGs in the root transcriptome by RNAseq analysis; (**b**) Chromosome-wise distribution of 3544 DEGs on 9 haploid sets of chromosomes of Carrot reference genome; up- (red dots) and down-regulated (blue dots) DEGs across Eastern and Western cv. are highlighted with different colors. (**c**) Volcano plot showing the up- (blue dots) and down-regulated (green dots) DEGs in Eastern and Western cv. of carrot, respectively.

**Figure 2 plants-12-03449-f002:**
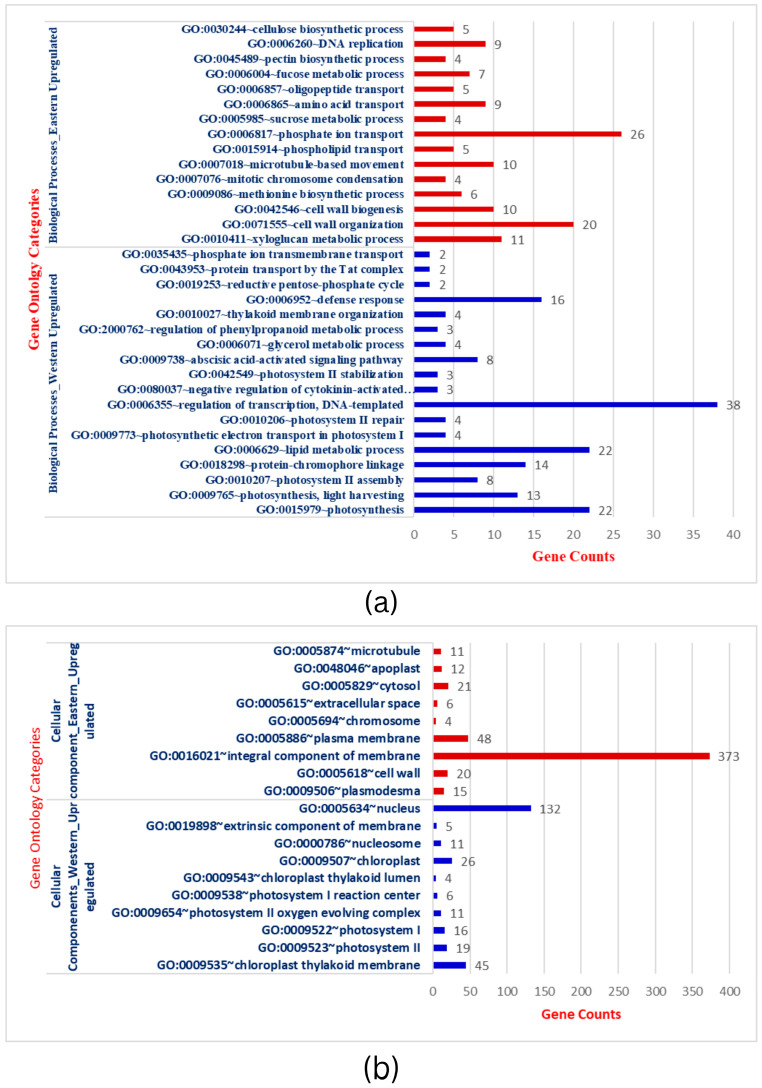
(**a**): Gene ontology categorization of Biological processes (BP) across Eastern and Western carrot cultivars and the number of genes showing respective BP. (**b**): Gene ontology categorization of Cellular Components (CC) across Eastern and Western carrot cultivars and the respective genes involved in CC. (**c**): Gene ontology categorization of Molecular Functions (MF) across Eastern and Western carrots and the respective number of genes involved in MF. (**d**): Comparison of DEGs involved in KEGG. Red bars indicate KEGG pathways upregulated in Eastern cv, while blue/green bars indicate KEGG pathways upregulated in Western cv.

**Figure 3 plants-12-03449-f003:**
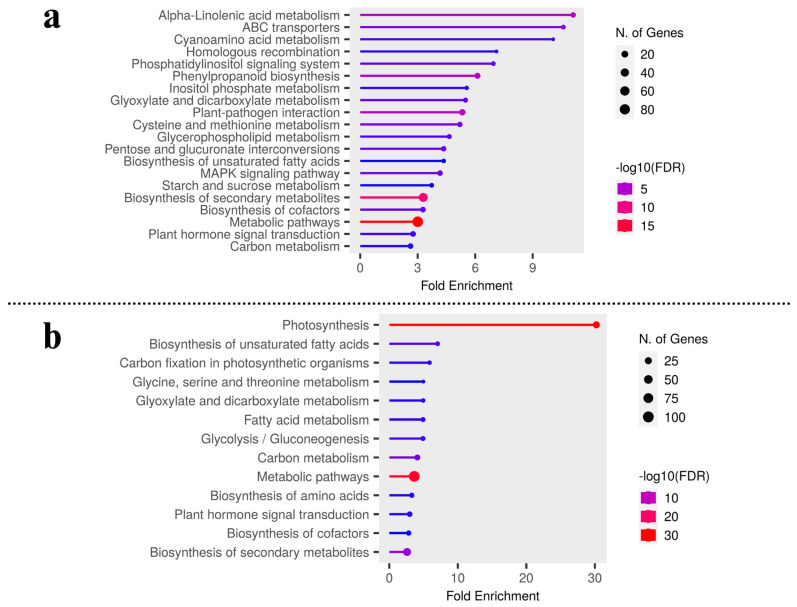
KEGG pathways enrichment analysis by gene set enrichment analysis in (**a**) Eastern cv. and (**b**) Western cv.

**Figure 4 plants-12-03449-f004:**
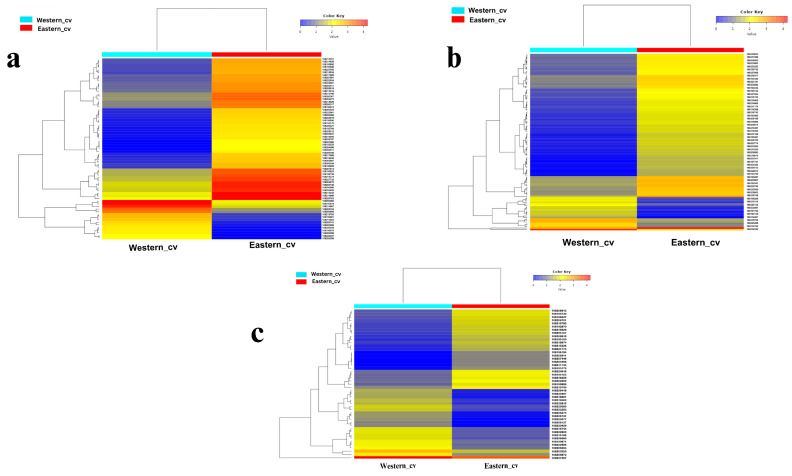
Heat map constructed based on Euclidean distance depicting the significant differentially expressed transcripts across Eastern and Western cvs. involved in (**a**) cell cycle and cell wall metabolism, (**b**) signal transduction, (**c**) starch and sucrose metabolism. Heatmaps are drawn based on FPKM values of significant DEGs across cvs.

**Figure 5 plants-12-03449-f005:**
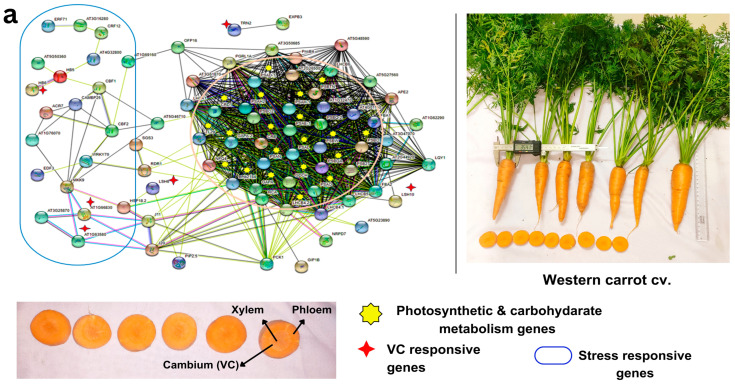
Gene regulatory network showing the interconnections of vascular cambium (VC) responsive genes and DEGs with stress-responsive genes and photosynthesis and carbon metabolism pathways in (**a**) Eastern carrot and (**b**) Western carrot based on an orthology search with *A. thaliana* and network drawn in STRING v. 11. Western and Eastern cv. root morphology and the vascular tissue (xylem, phloem and cambium and color) that are distinct across the cultivars are highlighted in the carrot vascular tissue.

**Figure 6 plants-12-03449-f006:**
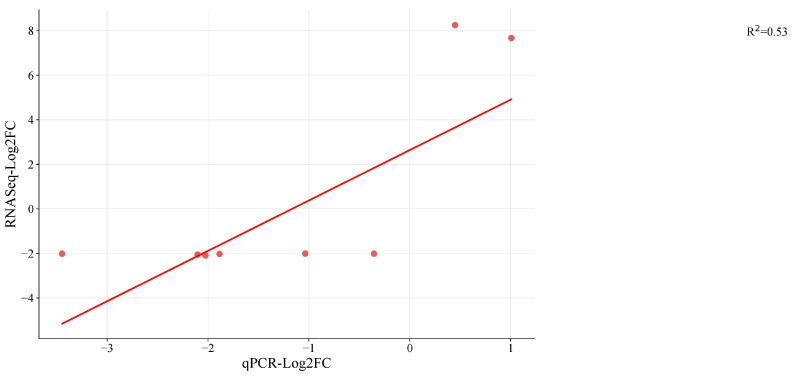
Validation of eastern vs. western DEGs identified in RNA seq by quantitative real-time PCR (qRT-PCR) in storage root transcriptome of cultivars of carrot by Log_2_FC. Red dots indicate the candidate genes used for RNAseq (Appendix A) the linear line is the regression line explaining qPCR and RNAseq results and R^2^ is the regression coefficient.

**Table 1 plants-12-03449-t001:** Morphological traits distinguishing Eastern and Western carrot cultivars.

Traits ^#^	Eastern Cultivar (UHSBC-23-1)	Western Cultivar (UHSBC-100)
Quantitative parameters *		
Plant height (cm)	68.13 ± 1.75 ^a^	53.76 0.73 ^b^
Root Length (cm)	16.52 ± 0.98 ^a^	13.18 ± 0.26 ^b^
Shoot length (cm)	52.49 ± 1.38 ^a^	46.64 ± 3.21 ^b^
Shoulder Width (cm)	21.53 ± 1.03 ^a^	16.22 ± 0.69 ^b^
Root diameter (cm)	26.07 ± 1.61 ^a^	25.39 ± 0.52 ^b^
Phloem width (mm)	50 ± 0.12 ^a^	85 ±0.14 ^b^
Xylem width (mm)	82 ± 0.08 ^a^	40 ± 0.01 ^b^
No. of petioles	7.91 ± 0.25 ^a^	8.46 ± 0.15 ^b^
Shoot weight (g)	70.91 ± 5.45 ^a^	24.49 ± 0.96 ^b^
Root weight (g)	72.70 ± 5.88 ^a^	40.66 ± 4.11 ^b^
Plot yield (100 plants) Kg	6.69 ± 0.62 ^a^	4.46 ± 0.34 ^b^
R/S ratio (Length)	0.32 ± 0.02	0.31 ± 0.01 ^b^
R/S ratio (Weight)	1.39 ± 0.15	1.83 ± 0.12 ^b^
TSS ° Brix	7.08 ± 0.11 ^a^	6.86 ± 0.14 ^b^
β-carotene mg/100 mg	6.5 ± 0.12 ^b^	8.5 ± 0.16 ^a^
Qualitative parameters		
Root Color (External)	Pale Orange/Red	Deep orange
Phloem color	Pale orange/Red	Deep orange
Xylem/core color	Yellow	Deep orange
Root texture	Coarse	Smooth
Root shape	Tapering	Cylindrical
Leaf type	Normal	Fern
Root hairiness	Moderate	Absent
Days to maturity	Early (80–85 days)	Late (90–95 DAS)
Flowering ability in the tropics	>80% Flowering	Absent

^#^ Observations were recorded from 100 individuals in both cvs. * Two samples, *t*-test assuming equal variance was used to test the significant difference between Eastern and Western cultivars for individual traits at *p* = 0.05. Similar letters indicate non-significant and different letters indicate the significant difference among the test cultivars.

**Table 2 plants-12-03449-t002:** Correlation pattern among the morphological characters in tested carrot cultivars.

Traits	NP	SL	PH	RL	PL	RWD	SWD	VW	XW	PW	RW
NP	1										
SL	0.055	1									
PH	0.282 **	0.058	1								
RL	0.165 *	0.182 *	0.273 **	1							
PL	0.15	−0.039	0.527 **	0.063	1						
RWD	0.18	−0.091	0.193 *	0.042	0.254 **	1					
SWD	0.084	0.049	0.691 **	0.069	0.507 **	0.558 **	1				
VW	0.726 **	−0.009	0.699 **	0.135	0.312 **	0.174 *	0.481 **	1			
XW	0.124	0.106	0.692 **	0.097	0.432 **	0.433 **	0.869 **	0.440 **	1		
PW	−0.178	−0.115	0.056	−0.059	0.200 *	0.440 **	0.450 **	−0.166 *	0.387 **	1	
RW	0.289 **	0.096	0.637 **	0.255 **	0.448 **	0.675 **	0.784 **	0.538 **	0.726 **	0.342 **	1

Note: NP = Number of petioles; SL = Shoot length (cm); PH = plant height (cm); RL = Root length (cm); PL = Petiole length (cm); RWD = Root width; SWD = Shoulder width; VW = Vegetative weight; XW = xylem width; PW = Phloem width; RW = Root weight. * Significance at *p* = 0.05; ** Significance at *p* = 0.01.

**Table 3 plants-12-03449-t003:** Summary of the cleaned reads of root transcriptome aligned to the reference genome by TopHat v2.0.6.

Sample Name	Left-Mapped Reads Rate			Right-Mapped Reads Rate			TopHat Mapping Rate (%)
	Input Reads	Total Mapped Reads	% Mapped Reads	Input Read	Total Mapped Reads	% Mapped Reads	
Western_R1	24,934,401	21,882,512	87.8	24,934,401	16,094,457	64.5	76.2
Western_R2	18,142,740	13,954,891	76.9	18,142,740	12,009,046	66.2	71.6
Western_R3	21,226,283	17,835,932	84.0	21,226,283	17,311,001	81.6	82.8
Eastern_R1	26,766,083	22,581,408	84.4	26,766,083	16,356,661	61.1	72.7
Eastern_R2	17,630,817	14,595,612	82.8	17,630,817	10,592,745	60.1	71.4
Eastern_R3	21,918,109	17,506,456	79.9	21,918,109	12,508,173	57.1	68.5

## Data Availability

The raw data is available at the NCBI database in the project ID PRJNA913450.

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
