# Peer review of "Comparative Root Transcriptome Profiling and Gene Regulatory Network Analysis between Eastern and Western Carrot (Daucus carota L.) Cultivars Reveals Candidate Genes for Vascular Tissue Patterning"

_plants, 2023, doi:10.3390/plants12193449_

Round 1
Reviewer 1 Report (Previous Reviewer 2)
Figure 4: The representation lacks resolution. The heatmap depicting genes and different genotypes is not clearly visible. Please ensure a higher resolution for better visibility.
Figure 6: Consider presenting this data as a regression plot instead of a bar plot and calculate the R-squared (R²) value for better analysis and visualization.
Need to work on grammar, english and sentence formation throughout manuscript.
Author Response
Response to Reviewer 1 Comments
- Summary
Thank you very much for taking the time to review this manuscript. Please find the detailed responses below and the corresponding revisions/corrections highlighted/in track changes in the re-submitted files.
- Point-by-point response to Comments and Suggestions for Authors
Comments 1: Figure 4: The representation lacks resolution. The heatmap depicting genes and different genotypes is not clearly visible. Please ensure a higher resolution for better visibility.
Response 1: We agree, accordingly improved the quality of figure4 as suggested
Comments 2: Figure 6: Consider presenting this data as a regression plot instead of a bar plot and calculate the R-squared (R²) value for better analysis and visualization.
Response 2: As suggested, Figure 6 is drawn with a regression plot with an R2 value. However, we assume the bar plot is more convincible and more commonly used in many other publications to show the validation of DEGs in qPCR. You may kindly retain the suitable plot for Figure 6.
Comment 3: Need to work on grammar, English, and sentence formation throughout the manuscript.
Response 3: We have enhanced the quality of the English grammar and expression of the manuscript by proofreading with a Native English-speaking colleague.
Thank you for your valuable critical review, and suggestions to improve the quality of the manuscript.
Reviewer 2 Report (Previous Reviewer 1)
The revised paper can be accepted.
Author Response
Response to Reviewer 2 Comments
- Summary
Thank you very much for taking the time to review this manuscript. Please find the detailed responses below and the corresponding revisions/corrections highlighted/in track changes in the re-submitted files.
- Point-by-point response to Comments and Suggestions for Authors
Comments 1: The revised paper can be accepted
Response 1: Thank you for your valuable review, suggestions, and kind acceptance.
This manuscript is a resubmission of an earlier submission. The following is a list of the peer review reports and author responses from that submission.
Round 1
Reviewer 1 Report
This study attempts to provide candidate genes for the vascular tissue configuration of Eastern and Western carrots through transcriptome analysis. Although differentially expressed genes were identified, no experiments have shown that this gene is related to vascular tissue development, as the differences between Eastern and Western carrots are not limited to vascular tissue.
In Figure 5, the author conducted qPCR to verify the accuracy of RNA-seq. One of the 9 genes is inconsistent with the RNA-seq results, although it is a minority, we cannot determine which result should prevail.
The font in Figure 1a is deformed, and the words in Figure 1b are almost unrecognizable.
In Figure 2d, it is more reasonable to label the columns corresponding to the two types of carrots with different colors.
Figure 3a and Figure 3b are overlapped.
The shooting angles of the Eastern Carrot and Western Carrot images in Figure 4 are inconsistent, and the arrangement of carrot slices is inconsistent; One of the two photos has a ruler and the other has no ruler.
Line number layout error: 286, 287, 308, 431, 456-476.
Insufficient innovation in this study and images are not standardized.
Reviewer 2 Report
There is no description of root plasticity and morphometric parameters measurement in carrot. In the table caption and method text, state how many replications were performed and also mentioned which was used in the method text.
How many normalized count genes were discovered (FPKM)? I'm surprised why the study only found 3544 DEGGs.
The DEGs associated with the Cell Cycle and Cell Wall Metabolism, signal transduction, and Starch and Sucrose Metabolism can be shown as a heatmap.
The reader does not understand the Gene regulatory network. WGCNA analysis can be performed on DEGS that have been identified as well as one of the vascular cambium patterning features.
The resolution of Figure 1 is really low.
A more detailed description of legends is required.
The English is weak and the sentence is extremely incomplete.